# Integrated Meta-Analysis Identifies Keratin Family Genes and Associated Genes as Key Biomarkers and Therapeutic Targets in Metastatic Cutaneous Melanoma

**DOI:** 10.3390/diagnostics15141770

**Published:** 2025-07-13

**Authors:** Sumaila Abubakari, Yeşim Aktürk Dizman, Filiz Karaman

**Affiliations:** 1Department of Statistics, Faculty of Arts and Science, Yildiz Technical University, Davutpaşa Campus, 34220 İstanbul, Türkiye; sumaila.abubakari@std.yildiz.edu.tr (S.A.); fkaraman@yildiz.edu.tr (F.K.); 2Department of Biology, Faculty of Arts and Sciences, Recep Tayyip Erdoğan University, 53100 Rize, Türkiye

**Keywords:** melanoma metastasis, druggability, bioinformatics, survival analysis, tumor microenvironment, protein-protein interaction

## Abstract

**Background/Objectives:** Cutaneous melanoma is one of the aggressive forms of skin cancer originating from melanocytes. The high incidence of melanoma metastasis continues to rise, partly due to the complex nature of the molecular mechanisms driving its progression. While melanomas generally arise from melanocytes, we investigated whether aberrant keratinocyte differentiation pathways—like cornified envelope formation—discriminate primary melanoma from metastatic melanoma, revealing novel biomarkers in progression. **Methods:** In the present study, we retrieved four datasets (GSE15605, GSE46517, GSE8401, and GSE7553) associated with primary and metastatic melanoma tissues and identified differentially expressed genes (DEGs). Thereafter, an integrated meta-analysis and functional enrichment analysis of the DEGs were performed to evaluate the molecular mechanisms involved in melanoma metastasis, such as immune cell deconvolution and protein-protein interaction (PPI) network construction. Hub genes were identified based on four topological methods, including ‘Betweenness’, ‘MCC’, ‘Degree’, and ‘Bottleneck’. We validated the findings using the TCGA-SKCM cohort. Drug-gene interactions were evaluated using the DGIdb, whereas structural druggability was assessed using the ProteinPlus and AlphaFold databases. **Results:** We identified a total of eleven hub genes associated with melanoma progression. These included members of the keratin gene family (e.g., KRT5, KRT6A, KRT6B, etc.). Except for the gene CDH1, all the hub genes were downregulated in metastatic melanoma tissues. From a prognostic perspective, these hub genes were associated with poor prognosis (i.e., unfavorable). Using the Human Protein Atlas (HPA), immunohistochemistry evaluation revealed mostly undetected levels in metastatic melanoma. Additionally, the cornified envelope formation was the most enriched pathway, with a gene ratio of 17/33. The tumor microenvironment (TME) of metastatic melanomas was predominantly enriched in NK cell–associated signatures. Finally, several hub genes demonstrated favorable druggable potential for immunotherapy. **Conclusions:** Through integrated meta-analysis, this study identifies transcriptional, immunological, and structural pathways to melanoma metastasis and highlights keratin family genes as promising biomarkers for therapeutic targeting.

## 1. Introduction

Skin cutaneous melanoma (SKCM) is an aggressive form of skin tumor originating from melanocytic cells. It accounts for the majority of skin-related mortalities, with an estimated five-year survival rate sharply dropping from 95–97.60% (localized) to under 20% in stage IV metastasis [1,2]. In 2025 alone, the American Cancer Society has estimated the incidence of melanoma at around 104,900 cases, comprising 60,500 in men and 44,400 in women [3]. Distributionally, a lifetime risk of being infected with melanoma has also been estimated at around 3% for Whites, 0.5% for Latinos/Hispanics, and 0.1% for Blacks, with exposure to ultraviolet (UV) identified as a major risk factor [4]. To this end, melanoma can be considered a major concern requiring timely intervention, both in early diagnosis and treatment [5,6]. Whereas early detection may lead to effective treatment through surgical excision, patients in the advanced stages tend to exhibit an unfavorable prognosis; hence, necessitating the identification of novel biomarkers underlying melanoma metastasis [7]. Thus, one way to improve patient prognosis may center around two complementary paths: embracing modern deep learning and computer-assisted image analysis—over the traditional physical dermatological examination—and screening novel biomarkers that underlie the cellular and molecular mechanisms of melanoma metastasis [8,9].

Like other cancers, melanoma progression is a function of time, influenced by several driving factors—epigenetic, genetic, or transcriptomic—that play vital roles in the tumor microenvironment (TME), such as tumor-infiltrating lymphocytes (TILs) [10]. In recent times, approaches adopted to minimize melanoma-associated mortalities include adoptive cell therapy (ACT), where the strategy is to extract TILs and reintroduce them to fight or eliminate the cancer [11,12]. In terms of treatment for advanced melanoma, a recent study of patients with resectable stage III melanoma showed a 1-year progression-free survival (PFS) of 83.7% for the neoadjuvant group treated with nivolumab + ipilimumab compared to the adjuvant group treated with only nivolumab immunotherapy [13]. Owing to the complex tumor microenvironment, both intrinsic and acquired drug resistance are still exhibited in more than 58% of patients with metastatic melanoma, and thus, significantly affecting the rate at which patients respond to immunotherapy [14,15,16]. Furthermore, the 31-gene expression profile (31-GEP) is another technique commonly used to estimate the risk of melanoma metastasis [5]. To this end, studying the molecular mechanisms underscoring melanoma progression and identifying novel biomarkers for targeted immunotherapies are vital to improving patients’ survival prospects.

The use of microarray technology and other high-throughput sequencing applications (e.g., high-throughput sequencing screening) has recently become the central focus of studying genetic alterations in the genome, ranging from cancer relapse and oncogenesis/tumorigenesis to novel biomarker screening [17,18]. For example, using three microarray datasets from the Gene Expression Omnibus (GEO), Li et al. [19] identified genes potentially linked to melanoma metastasis through keratinocyte differentiation, concluding that such genes latently provide molecular insights into melanoma progression. However, the study did not explore the role of these metastasis-associated genes in immune infiltration. In a similar study, Luan et al. [20] also identified hub genes for cutaneous melanoma, primarily focusing on metastasis-associated genes and their potential interplay in the tumor microenvironment (TME). While these studies and others have linked keratin family genes to melanoma metastasis, their functional relevance to immune cell infiltration and their therapeutic potential remain underexplored.

In the present study, we use bulk RNA expression profiles from Gene Expression Omnibus (GEO), including GSE15605, GSE46517, GSE8401, and GSE7553, to compare the expression of metastatic melanoma and primary melanoma tissues. Here, we screened differentially expressed genes (DEGs) to analyze the molecular mechanisms underlying melanoma metastasis through Kyoto Encyclopedia of Genes and Genomes (KEGG), Gene Ontology (GO), and protein-protein interaction (PPI) networking. The DEGs identified were based on meta-analysis using the DExMA package and validated the expression levels of the screened genes using the Cancer Genome Atlas (TCGA-SKCM), including clinical samples. Additionally, the hub genes obtained from the PPI network analysis were analyzed with the TIMER web-based tool to estimate their immune cell infiltration. To underscore potential therapeutic targets of the hub genes. Finally, we use the DGIdb database to identify drug-gene interactions of the candidate biomarkers and assess their druggability potential using the ProteinPlus (https://proteins.plus/, accessed on 15 June 2025) and AlphaFold (https://alphafold.ebi.ac.uk/, accessed on 15 June 2025) repositories. This potentially lays the groundwork for therapeutic repurposing and immune-related medicine in melanoma metastasis [21,22].

## 2. Materials and Methods

### 2.1. Data Selection and Preprocessing (Inclusion-Exclusion Criteria)

Using the GEO database, we performed a systematic search focusing on keywords such as “cutaneous melanoma”, “primary melanoma”, “metastatic melanoma”, “gene expression”, “gene expression profiling” and “*Homo sapiens*”. This resulted in a total of 142 search results. Three independent researchers were tasked to carefully review the titles and abstracts of the studies so as to determine what to include or exclude. The inclusion criteria were; i. must include both melanoma and metastatic melanoma samples; ii. sample size per group not less than five (5); iii. skin-derived tumor tissues; and iv. platform consistency. As presented in Table 1, we settled on four datasets, including GSE7553 (https://www.ncbi.nlm.nih.gov/geo/query/acc.cgi?acc=GSE7553, accessed on 10 May 2025) [23], GSE15605 (https://www.ncbi.nlm.nih.gov/geo/query/acc.cgi?acc=GSE15605, accessed on 10 May 2025) [24], GSE46517 (https://www.ncbi.nlm.nih.gov/geo/query/acc.cgi?acc=GSE46517, accessed on 10 May 2025) [25], and GSE8401 (https://www.ncbi.nlm.nih.gov/geo/query/acc.cgi?acc=GSE8401, accessed on 10 May 2025) [26]. The GEOquery R package (version 2.76.0) was used to download these datasets. The datasets GSE15605 and GSE7553 were based on the GPL570 platform, while the datasets GSE8401 and GSE46517 were based on GPL96. Additionally, the dataset GSE7553 included basal cell carcinoma and squamous cell carcinoma samples, which were accordingly excluded in the downstream analysis.

### 2.2. Quality Control (QC) and Batch-Effect Correction

Following the selection of the four datasets, including the expression matrix, clinical, and annotation data, we used the limma and edgeR R packages to process the data further. Except for the GSE15605 data, the expression profiles of the other three datasets were log-transformed (i.e., $\log_{2}\left(a+1\right)$). The normalizeBetweenArrays function was used to perform quantile normalization, whereas the filterByExpr function was used to filter out lowly expressed genes. For multiple probes mapped to a single gene, we used the mean statistic to quantify their expression level. The probes were subsequently mapped to their corresponding HUGO Gene Nomenclature Committee (HGNC) names prior to the batch effect correction process.

Although combining multiple batches of genomic data increases statistical power—in what is normally termed cross-platform harmonization—the approach risks introducing systematic biases due, for example, to varying sequencing platforms or variability due to probe design [27]. To overcome this problem, batch-effect correction techniques are often applied. In this study, we used the sva function from the ComBat package to correct platform-based bias (i.e., GPL96 vs. GPL570). To assess the effectiveness of the correction, we obtained a visualization of the ‘before’ and ‘after’ correction (based on principal component analysis). The resulting ‘combined’ dataset was used in the downstream analysis of melanoma metastasis.

### 2.3. Tumor Purity Filtering

To minimize noise from non-tumor samples across the four datasets, we applied the *ESTIMATE* algorithm (Estimation of STromal and Immune cells in MAlignant Tumor tissues using Expression) to estimate tumor purity [28]. Thus, using the estimate R package, stromal and immune scores were computed, and a tumor purity threshold of ≥75% was set to maintain high-confidence tumor samples. This critical step improves differential expression analysis by minimizing confounding immune and stromal influences. Additionally, the setting of method = epic in the immunedeconv R package was used to deconvolve immune cell fractions for the downstream analyses.

### 2.4. Integrated Meta-Analysis of Differentially Expressed Genes

Following quantile normalization and filtering of lowly expressed genes, the limma and edgeR R packages were used to perform differential gene expression (DEGs) between primary and metastatic melanoma tissues. For each of the four datasets, the criteria to identify expressed genes were the logFoldChange (i.e., |logFC|>1.50) and the adjusted *p*-value (i.e., adj.P.Val<0.05).

As previously mentioned, platform differences tend to introduce systematic bias. To overcome the bias, batch-effect correction—via the ComBat R package—is commonly used to harmonize the varying datasets. In this case, we used the ‘pooled’ dataset to identify differentially expressed genes associated with melanoma metastasis. Furthermore, enhancing robustness and statistical power can be achieved through well-established methods to identify differentially expressed genes. In this study, we used two R packages, namely RobustRankAggreg (RRA) and DExMA, to complement the batch-corrected dataset. The RRA uses the aggregateRanks function to rank differentially expressed genes across multiple datasets by integrating and prioritizing consistently high-ranked genes. In contrast, the metaAnalysisDE in DExMA uses a fully fledged meta-analysis pipeline to perform differential expression using the raw or normalized expression data [29]. In this study, we used Fisher’s method to identify DEGs.

### 2.5. Immune Cell Infiltration Analysis

Quantifying immune cell proportions (e.g., macrophages, CD8+, etc.) in the tumor microenvironment is an important path to understanding tumor progression [30]. Using the immunedeconv R package, we correlated identified expressed genes (|log2FC|>1.5, FDR<0.05) with immune cell fractions estimated by the EPIC deconvolution method. In this study, we focused on the correlations between the estimated fractions of B cells, CD4+/CD8+ T cells, NK cells, macrophages, cancer-associated fibroblasts (CAFs), and endothelial cells and hub genes. Correlation coefficients were computed using the Spearman correlation method, with, *p* values <0.05 considered as statistically significant. Using the pheatmap and ggplot2 packages, heatmaps and correlation plots were used to characterize the proportion of immune infiltrates between primary and metastatic melanoma tissues.

### 2.6. Functional Enrichment Analysis of DEGs

We used the clusterProfiler and enrichplot R packages to perform enrichment analysis to identify significantly enriched biological processes and molecular functions underlying the differentially expressed genes. For functional annotation, we queried the Gene Ontology (GO) and Kyoto Encyclopedia of Genes and Genomes (KEGG) databases with well-established curated repositories on gene functions, processes, cellular locations, and biological pathways across different species. Our study mainly focused on biological process (BP), cellular component (CC), and molecular function (MF) categories for the gene ontology (GO) analysis of the DEGs. The Benjamini-Hochberg method (i.e., false discovery rate) was used to evaluate statistical significance (*p*.val.adj < 0.05).

### 2.7. Protein-Protein Interaction Network and Hub Gene Identification

For functional associations and hub genes screening, the STRING database (https://string-db.org/cgi/input?sessionId=baTnyK0lJLgM&input_page_show_search=on, accessed on 18 May 2025) is one of the most comprehensive resources for identifying protein-protein interactions (PPI) in biological systems. In the current study, we constructed PPIs using the DEGs obtained from the integrated meta-analysis. We used an interaction threshold score of least 0.70 (i.e.,Iscore≥0.70) for significance. The resulting PPI network was visualized using Cytoscape software (https://cytoscape.org/, accessed on 18 May 2025) (version 3.10.3). Furthermore, we downloaded the Cytoscape’s CytoHubba plugin for a topological approach to identifying nodes. To this end, we used five ranking methods in Cytoscape, namely “MCC”, “Betweenness”, “Degree”, and “Bottleneck”. By prioritizing the top 20 ranked genes, hub genes were identified as the one overlapping all the five methods. Initially, weights for the five methods were treated equally and repeated again with weighted rank aggregation (following score normalizaton, i.e., $x_{\mathrm{norm}}=x$/max(x)).

### 2.8. Validation and Prognostic Assessment of Hub Genes (TCGA-SKCM)

We used the UALCAN web portal (https://ualcan.path.uab.edu/index.html, accessed on 23 June 2025) to validate the expression levels (mRNA) of the hub genes. UALCAN is a free, user-friendly tool used by researchers to analyze omics data from TCGA, CPTAC, etc., where we can compare tumors (primary or metastatic) vs. normal tissues as well as stratify survivability across varying risk factors. A *p*-value threshold (i.e., p<0.05) was set for statistical significance. Using median expression levels across the dataset, we stratified expression levels into high- and low-expressed genes, followed by evaluating the overall survival (OS) time between melanoma and metastatic melanoma. To evaluate protein expression across normal and cancer tissues, immunohistochemistry (IHC) examination was performed using the Human Protein Atlas (HPA) web tool (https://www.proteinatlas.org/, accessed on 17 June 2025).

### 2.9. Drug–Gene Interaction and Structural Druggability Analysis

For translational goals, we used the interactive DGIdb web-based database (https://dgidb.org, accessed on 20 May 2025) to identify potential therapeutic targets among the expressed hub genes [31]. Thus, we first queried drug-gene interactions by downloading the interaction score, regulatory approval status, and drugs likely linked to the identified hub genes. We used the Cytoscape again to build a network of drug-gene interactions. The AlphaFold rotein structure database (https://alphafold.ebi.ac.uk/, accessed on 15 June 2025) [22] was used to download predicted protein structures (.pdb files) for molecular docking in ProteinPlus (https://proteins.plus/, accessed on 15 June 2025) [21]. We used the resulting PDB files on ProteinsPlus to assess the binding pocket volume and druggability score (i.e., 0≤score≤1). A druggability score higher than 0.6 was considered ‘high-confidence’ for structure-based design. We provided a table summarizing the druggability score of the hub genes as potential therapeutic targets in metastatic melanoma.

## 3. Results

### 3.1. Data Selection, Quality Control, and Tumor Purity Filtering

Following a careful review of GEO datasets on primary melanoma and metastatic melanoma tissues, a total of 4 datasets were retained for the downstream analysis, including GSE15605, GSE7553, GSE8401, and GSE45617, as shown in Table 1. To ensure data integrity for GSE7553, samples of basal cell carcinoma (BCC; 15), squamous cell carcinoma (SCC; 11), melanoma in situ (2), normal (4), and normal epidermis (1) were excluded in the downstream analysis. Because each dataset was either GPL570- or GPL96-platformed, we performed batch-effect correction to ensure a cross-platform harmonized dataset. The effectiveness of the correction was assessed with a PCA visualization (Figure 1A). Before correction, the results showed PC1 of ∼48% and PC2 of ∼20%. Thus, nearly 70% of total variation was ‘locked’ up in two directions. Post-batch-effect correction, the variation dropped to ∼35% and ∼16%, respectively. The shift indicates a redistribution of variation across the datasets, ensuring less distinct clusters between the samples (GPL570 vs. GPL96). Similarly, using ESTIMATE in R, stromal and immune scores were computed to evaluate tumor purity of the samples (Figure 1B) with a purity threshold set at 75%. Low-purity tumors tend to have high stromal/immune scores, while high-purity tumors are more “tumor-cell-dominant”.

### 3.2. Integrated Meta-Analysis of Differentially Expressed Genes

A total of 356 DEGs, including 312 upregulated and 44 downregulated genes, were identified in GSE15605. In GSE46517, a total of 164 DEGs, including 161 upregulated and 3 downregulated genes; a total of 241 DEGs, including 219 upregulated and 22 downregulated genes in GSE7553; and a total of 242, including 229 upregulated and 13 downregulated genes in GSE8401 (Figure 2A–D). Overall, 80 metastasis-associated genes were shown to overlap all four datasets (Figure 3A). Comparatively, the results in meta-analysis using the ‘pooled’ (i.e., batch-corrected), rank (i.e., RobustRankAggreg), and DEXMA methods showed 145 DEGs overlapping the RRA and DEXMA methods, whereas 17 DEGs overlapped all three methods (Figure 3B).

### 3.3. Immune Cell Infiltration Analysis

We used the EPIC algorithm via the immunedeconv R package to profile the degree of immune cell infiltration between primary and metastatic melanoma tissues. As shown in Figure 4, the results revealed diverse range of immune cell types across the tumor classes—primary melanoma and metastatic melanoma tissues. Observably, the primary melanomas revealed more heterogeneous immune cell composition, with comparatively greater variability in CD8+/CD4+ T cells and macrophages. On the contrary, metastatic melanoma samples showed a relatively uniform infiltration pattern, with consistently high NK cell proportions, and in some cases, too, increased endothelial cell or B cell fractions. Moreover, the CD8+ T cell abundance was relatively reduced in metastatic samples compared to primary melanoma tissues.

### 3.4. Functional Enrichment Analysis (KEGG/GO)

Using the clusterProfiler and enrichplot packages, we performed functional enrichment analysis of the DEGs via the GO/KEGG complementary databases. In the gene ontology analysis, we primarily focused on the biological processes (BP), cellular component (CC), and molecular function (MF), with the false discovery rate evaluated with the Benjamini-Hochberg method. The results revealed biological processes mainly enriched in “skin development”, “keratinocyte differentiation”, “epidermal cell differentiation”, “epidermis development”, etc. (Figure 5A). Moreover, molecular function showed enrichment in “structural constituent of skin epidermis”, “structural constituent of cytoskeleton”, “RAGE receptor binding”, etc. (Figure 5B), whereas cellular components highlighted enriched components such as “cornified envelope”, “intermediate filament cytoskeleton”, “intermediate filament”, “keratin filament”, etc. (Figure 5C). In terms of the KEGG pathways, we found “cornified envelope formation” (*hsa04382*) as the most enriched pathway (FoldEnrichment: 22.3; GeneRatio: 17/33; p.adjust: 5.89 $\times 10^{-18}$), as shown in Figure 6. The 17 genes found in this pathway and the enrichment map of the Go terms (Figure 5) are shown in the Appendix A.

### 3.5. Protein-Protein Network and Hub Gene Identification

To identify hub genes, the DEGs via meta-analysis were fed into the STRING interactive web tool (https://string-db.org/, accessed on 18 May 2025) necessary for the construction of the PPI network. The minimal threshold of interaction confidence for the network was set at 0.70 (i.e.,Iscore≥0.70). The resulting protein-protein interactions, including the number of nodes, edges, and enrichment, are presented at the metastatic melanoma PPI network (https://version-12-0.string-db.org/cgi/network?networkId=bdUMLtVoOHNE, accessed on 18 May 2025). Using CytoHubba, the top 30 leading genes, first with equal weights and second with rank aggregation, were identified using four different ranking methods, namely, “MCC”, “Betweenness”, “Degree”, and “Bottleneck” (Figure 7A–D). The result revealed eleven hub genes from the top 20 overlapping genes (Figure 7B,C), including *KRT5*, *FLG*, *KRT6A*, *KRT16*, *KRT6B*, *KRT10*, *DSP*, *SPRR1B*, *PI3*, *CDH1*, and *S100A7*.

### 3.6. Validation and Prognostic Assessment of Hub Genes (TCGA-SKCM)

Using the TCGA-SKCM dataset comprising 368 melanoma metastases and 104 primary melanoma tissues, we validated the mRNA expression of the eleven (11) hub genes. We compared the expression levels between primary and metastatic melanoma levels using the wilcoxon.test function in R with statistical significance set at *p*-value <0.05. The results, as shown in Figure 8A, revealed significant decreased expression levels in metastatic melanoma, except for the gene CDH1. Based on the hazard ratios (HR), we also evaluated the overall survival prospects of the hub genes at the univariate level, classifying each hub gene as either favorable (i.e., if HR<1) or unfavorable (i.e., if HR>1). The results showed all the hub genes having unfavorable survival prospects (Figure 8). Furthermore, we used the Human Protein Atlas database (HPA) (https://www.proteinatlas.org/, accessed on 17 June 2025) to assess the protein expression of the hub genes by checking their immunohistochemistry (IHC) status (Figure 9). The results showed varying staining levels between the primary and metastatic melanoma tissues.

### 3.7. Drug-Gene Network Analysis

The main focus of performing drug-gene interaction and druggability analyses is to draw insights into translational medicine, from molecular discovery to potential therapy. We first used the DGIdb database to collect data on interaction score, regulatory approval, and potential therapeutic targets of the hub genes. Using the DGIdb, we identified five unique drug-gene interactions of the hub genes, involving *KRT10*, *DSP*, *CDH1*, *FLG*, and *PI3*, with three of these genes, namely *DSP* (ENALAPRIL MALEATE, NINTEDANIB ESYLATE), *CDH1* (BICALUTAMIDE), and *PI3* (PROGESTERONE), currently approved. Using Cytoscape, a drug-gene interaction network was constructed, revealing multiple interactions, as shown in Figure 10. The interaction scores and approval statuses of the hub genes are summarized in Table 2. At the structural level, the predicted protein structures of the eleven hub genes were downloaded from the AlphaFold Protein Structure Database (https://alphafold.ebi.ac.uk/, accessed on 15 June 2025) [22]. To this end, the druggability potential of the hub genes using ProteinPlus (https://proteins.plus/, accessed on 15 June 2025) [21] was estimated with the DogSiteScore. Based on a DogSiteScore of at least 0.60, the results revealed eleven hub genes—including CDH1, DSP, and several keratins—with favorable structural properties for druggability (Table 3).

## 4. Discussion

Cutaneous melanoma is the most widely known form of skin tumor, accounting for most skin-associated deaths, primarily due to its aggressiveness [1,33]. If it is not detected early and attended to, it can quickly metastasize to other body parts, leading to poor patient survival prospects. Therefore, a comprehensive bioinformatics study underscoring the cellular and/or molecular mechanisms of metastatic melanoma progression is important for targeted therapy. In the last 2/3 years, research on skin cutaneous melanoma has intensely focused on several paths, from artificial intelligence in diagnosis [8] and emerging biomarkers and metastatic reactivation [34] to melanoma microenvironment and aging. In several studies (e.g., Xie et al. [35]) involving biomarker identification of melanoma metastasis, relying on single datasets often yields findings with limited generalizability, largely due to the absence of cross-cohort data integration and harmonization. Moreover, many studies do not explore the druggability of identified hub genes or consider the modulatory role of immune cell infiltration.

In the present study, our analysis identified 80 differentially expressed genes linked to cutaneous melanoma metastasis. Of this number, 17 genes were consistently identified across all three methods (i.e., pooled, RRA, and DExMA), underscoring a conserved signature linked to melanoma metastasis. Furthermore, the predominance of upregulated genes gives a hint of a potential activation of metastasis-linked pathways. In the functional analysis, gene ontology analysis identified significantly enriched terms, including “epidermis development”, “skin development”, “keratinocyte differentiation”, “cornified envelope”, “intermediate filament”, “desmosome”, “structural constituent of skin epidermis”, “structural constituent of cytoskeleton”, and “endopeptidase inhibitor activity”, whereas the KEGG pathway enrichment analysis of the DEGs primarily showed high enrichment in the “cornified envelope formation” pathway. These findings suggest aberrant keratin expression may indicate disrupted epithelial architecture or altered differentiation states linked to metastatic behavior. At the cellular level, however, “intermediate filament” suggests cytoskeletal remodeling, enhancing invasion. Past studies have reported the potential roles of “desmosome”, “epidermis development”, “intermediate filament”, “cornified envelope,” and “keratinocyte differentiation” in melanoma metastasis, including BRAF resistance, plasticity, immune surveillance evasion, and promotion of melanoma metastasis through oxidative stress conditioning [36,37,38,39].

As mentioned earlier, research on melanoma metastasis using transcriptomic profiling has expanded considerably in recent years, including the identification of epigenetically modulated genes. Multiple hub genes (including the keratin family) have been implicated in many cancer types in their roles in tumor progression, immune evasion, and epithelial differentiation. KRT5/KRT18, for example, are dysregulated in cancers originating from basal epithelial cells, playing a role in epithelial-mesenchymal transition (EMT) in basal-like cancers such as breast cancer [40]. The direct roles of desmoplakin (DSP) and peptidase inhibitor 3 (PI3) in melanoma metastasis are largely limited, though they are generally involved in cell-cell adhesion. However, the dysregulation of these genes in various cancers has, in past studies, been linked to tumor progression. For example, Hao et al. [41] identified SOX30 as a core regulator of a tumor-suppressive desmosomal gene in lung adenocarcinoma. Similarly, Harada et al. [42] identified the dysregulation of PI3 linked to gastric cancer. Therefore, we argue that while further studies are required to throw more light on the roles of these genes in melanoma metastasis, the molecular mechanisms underlying their roles in other cancers highlight their potential as biomarkers in targeted therapies.

In our KEGG analysis, we found the most significant enriched pathway to be “cornified envelope formation” (GeneRatio = 17/33, *p*.adjus = 5.89 $\times 10^{-18}$). The genes found in this pathway included SPRRs, LOR, IVL, FLG, KRT6A, etc. The cornified envelope (CE) is a protein-lipid structure extremely vital for skin barrier activity owing to its formation of mechanical barriers. Thus, it replaces the plasma membrane of differentiated keratinocytes [43]. As Candi et al. [43] noted, the cornified envelope is a transglutaminase-crosslinked envelope associated with terminal epidermal differentiation. Therefore, this highly enriched pathway may reflect ectopic expression of skin differentiation markers by melanoma cells, epidermal contamination in the bulk RNA, or inflammation-associated reprogramming (via SPRRS). Moreover, cornification and barrier-related processes may contribute to immune modulation and metastatic potential. However, to lend further evidence to this observation, we may need further bioinformatics study involving spatial transcriptomics or single-cell RNA sequencing.

At the validation level of gene expression (TCGA-SKCM), except for CDH1, all the hub genes were downregulated in metastasis as compared to primary melanoma tumors (Figure 8A). Lower keratin expression in metastases may reflect reduced epithelial or keratinocyte content and increased stromal or immune infiltration, suggesting a shift in tumor microenvironment composition rather than intrinsic tumor cell changes. Keratin genes are generally canonical markers; hence, their lower expression in metastasis reflects reduced epithelial or keratinocyte composition and increased immune or stromal infiltration. This may point to a shift in the TME constitution. Oddly, the upregulation of CDH1 in metastatic tissues, while counterintuitive, may reflect heterogeneity of gene expression across melanoma tissues or may suggest plasticity during metastatic melanoma progression, thus requiring further validation [44]. Furthermore, protein expression level validation through immunohistochemistry (IHC) staining showed mostly ‘undetected’ (except CDH1) in the metastatic samples.

Our study also considered immune cell infiltration in the TME—an approach aimed at revealing the cellular composition of cell infiltrates—including factors influencing tumor biology, therapy resistance, and patient’s overall survival [30]. In our study, using EPIC for immune deconvolution between primary and metastatic melanoma revealed a more heterogenous immune environment, especially in CD4+/CD8+ T cells and macrophages, reflecting varying immune surveillance. Additionally, metastatic tumors in the TME were predominantly in the NK cells and, to some extent, increased endothelial or B cells. Inferentially, the reduced volume of CD8+ T cells in the metastatic tissues underscores immune invasion, a factor that may contribute to poor prognosis. This observation was further validated with the latest TIMER3 web tool (http://timer.cistrome.org/, accessed on 21 June 2025) (based on B cells, CD8+/CD4+ T cells, macrophages, neutrophils, and dendritic cells).

In our study, the drug-gene interaction analysis of the 11 hub genes revealed genes such as PI3 (downregulated), DSP (downregulated), and CDH1 (upregulated), underscoring translational potential. Though PI3 interacts with progesterone, its relevance as an anti-cancer agent in melanoma remains unclear. Furthermore, DSP interacted with approved drugs such as enalapril and nintedanib. Although their roles in melanoma metastasis are unclear, they may warrant investigation for therapeutic repurposing. Presently, targeted therapies for the treatment of metastatic melanoma are stratified based on tumor stage (e.g., stage IV), mutant genotype (e.g., MEK/BRAF inhibitors), or the expression levels of key biomarkers such as PD-L1 [45,46]. Of the keratin family genes, KRT10 was the most promising for therapeutic epidermal treatment. Based on the AlphaFold and druggability score estimation, the results suggested that keratins like KRT16 and FLG are structurally favorable drug targets. While these integrative findings appear to suggest their potential for future targeted therapies, experimental validation, including docking or CRISPR-based approaches, would be needed for targeted actions.

This study has several limitations. Primarily, the use of bulk transcriptomic data inherently masks the cellular heterogeneity within melanoma tumors. Given the plasticity of melanoma and its complex microenvironment, differential gene expression may reflect shifts in cell composition rather than true transcriptional changes in malignant cells. Moreover, melanoma consists of distinct molecular subtypes (e.g., BRAF-mutant, NRAS-mutant, triple wild-type), each with unique gene expression profiles. Without stratification by subtype, some differentially expressed genes would confound inference. While our integrated meta-analysis appears to enhance robustness, it cannot fully resolve these molecular complexities. A possible future direction might be employing integrative analyses using single-cell or spatial transcriptomic profiling methods that would better capture intratumoral heterogeneity.

## 5. Conclusions

In this study, through extensive bioinformatics and meta-analysis, we identified 80 DEGs associated with the transition from primary melanoma to metastatic cutaneous melanoma, revealing 11 potential melanoma-associated hub genes—mostly keratins. From a prognostic perspective, downregulation in metastatic melanoma tissues of these hub genes (KRT5, KRT6A, KRT6B, KRT10, KRT16, FLG, DSP, SPRR1B, PI3, and S100A7) revealed an unfavorable prognosis. The KEGG analysis of the hub genes revealed strong significance in “cornified envelope formation”, a keratinocyte-linked pathway suggesting epithelial reprogramming in the melanoma TME. Moreover, immune cell deconvolution revealed a reduction of CD8+/CD4+ T cell infiltration in melanoma progression. Using drug-gene interaction results from the DGIdb database, the AlphaFold and ProteinPlus web-based tools were used to estimate druggability score, with the results suggesting that keratins like KRT16 and FLG are structurally favorable drug targets. While these integrative findings appear to suggest their potential for future targeted therapies, experimental validation, including docking or CRISPR-based approaches, would be needed for targeted actions.

## Figures and Tables

**Figure 1 diagnostics-15-01770-f001:**
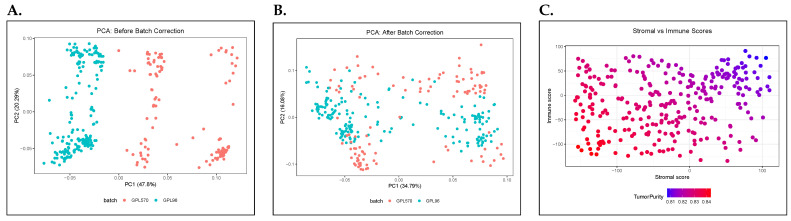
Quality control, batch-effect correction, and tumor purity filtering. (**A**) Principal Component Analysis (PCA) plot before batch-effect correction between GSE15605, GSE7553, GSE8401, and GSE45617 datasets. (**B**) PCA plot post batch-effect correction. (**C**) Tumor purity distribution following ESTIMATE-based filtration.

**Figure 2 diagnostics-15-01770-f002:**
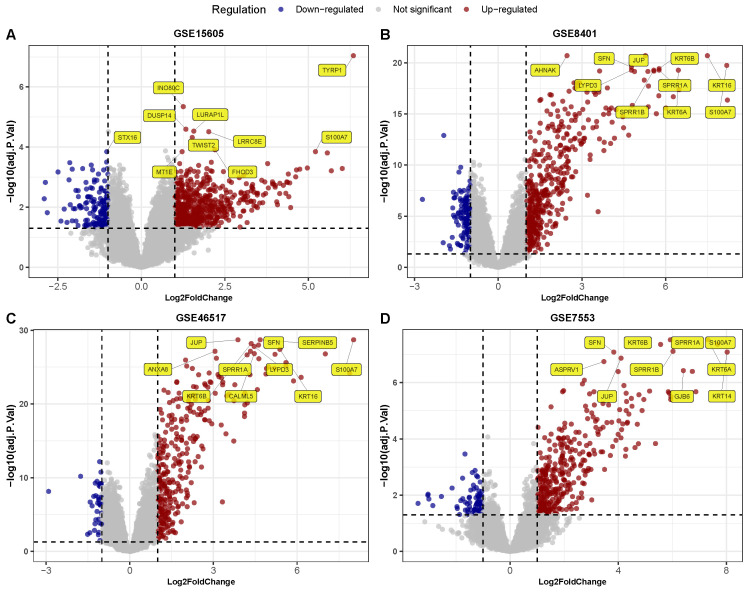
Integrated DEGs and meta-analysis of melanoma metastasis. (**A**–**D**) Volcano plots across 4 GEO datasets, showing key up- and downregulated genes. The top 10 highly expressed are highlighted in yellow. The DEGs were identified using the criteria |logFC|>1.50;FDR<0.05.

**Figure 3 diagnostics-15-01770-f003:**
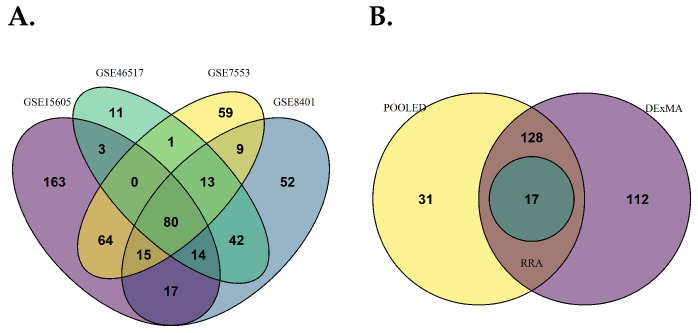
Identification of overlapping DEGs. (**A**) Venn diagram showing overlapping differentially expressed genes as shared transcriptional signatures. (**B**) Overlap of robust DEGs identified through meta-analysis using three approaches: pooled analysis (batch-corrected), Robust Rank Aggregation (RRA), and DExMA, revealing consistently dysregulated genes across methodologies.

**Figure 4 diagnostics-15-01770-f004:**
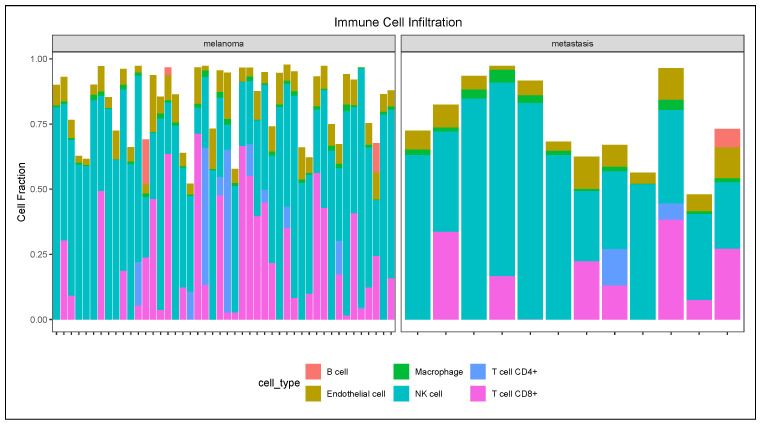
Immune cell infiltration pattern between primary and metastatic melanoma. Inferred immune cell abundance using the EPIC algorithm in the ‘immunedeconv’ R package.

**Figure 5 diagnostics-15-01770-f005:**
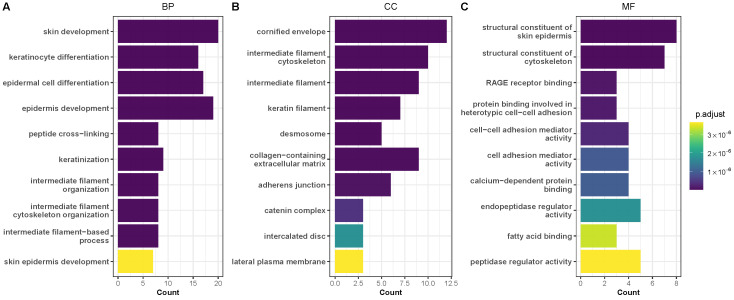
Functional enrichment analysis of genes associated with melanoma metastasis (cutaneous). (**A**–**C**) Barplot of GO enrichment analysis based on biological processes, cellular component, and molecular function. The gradient bar (light to dark) indicates adjusted *p*-value using the Benjamini-Hochberg method. ‘Count’ on the horizontal axis indicates the number of genes contributing to enrichment. GO: gene ontology; BP: biological process; MF: molecular function; CC: cellular component; DEGs: differentially expressed genes.

**Figure 6 diagnostics-15-01770-f006:**
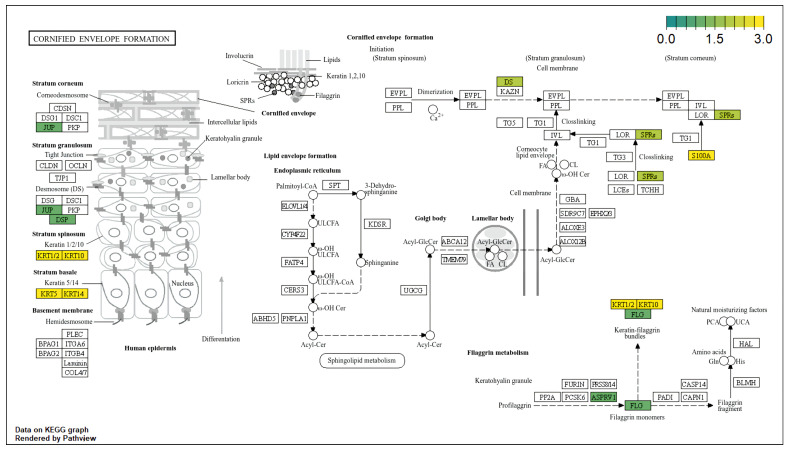
KEGG pathway changes of DEGs. Genes significantly expressed in the “cornified envelope formation” pathway. Color gradients highlight gene expression changes in metastatic vs. primary melanoma tissues (visualized using the pathview R package by Luo et al. [32]).

**Figure 7 diagnostics-15-01770-f007:**
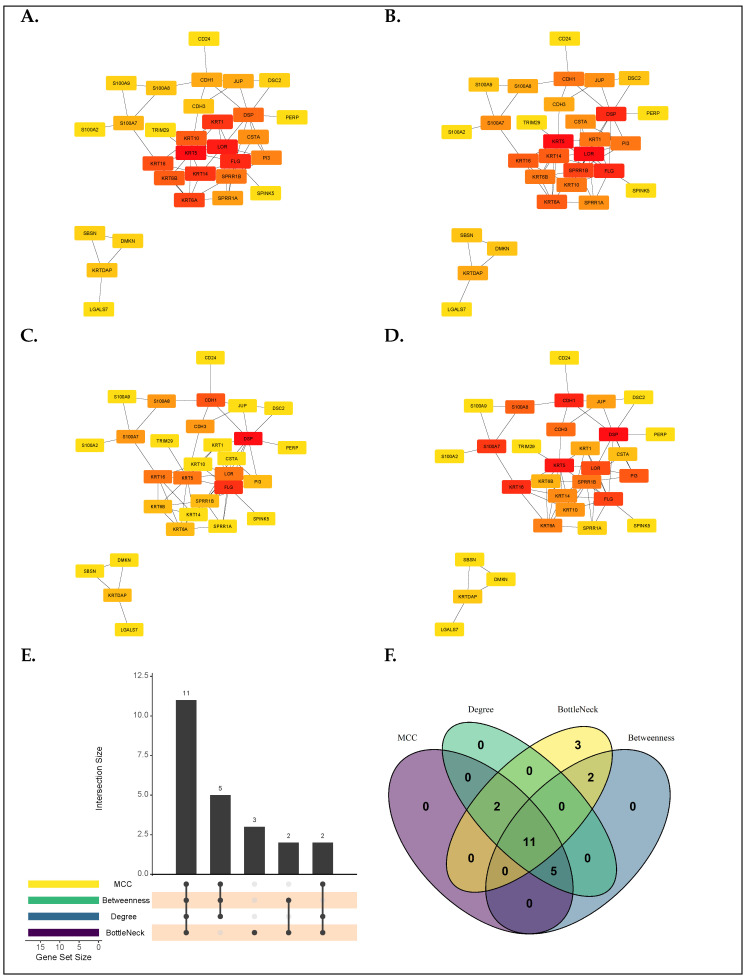
Identification of hub genes via PPI network. (**A**–**D**) Hub genes were identified using four topological algorithms in cytoHubba, namely (**A**). MCC (**B**). Degree (**C**). BottleNeck (**D**). Betweenness. (**E**) Identifying top-ranked genes via rank aggregation (**F**) Identifying top-ranked genes with equally weighted methods.

**Figure 8 diagnostics-15-01770-f008:**
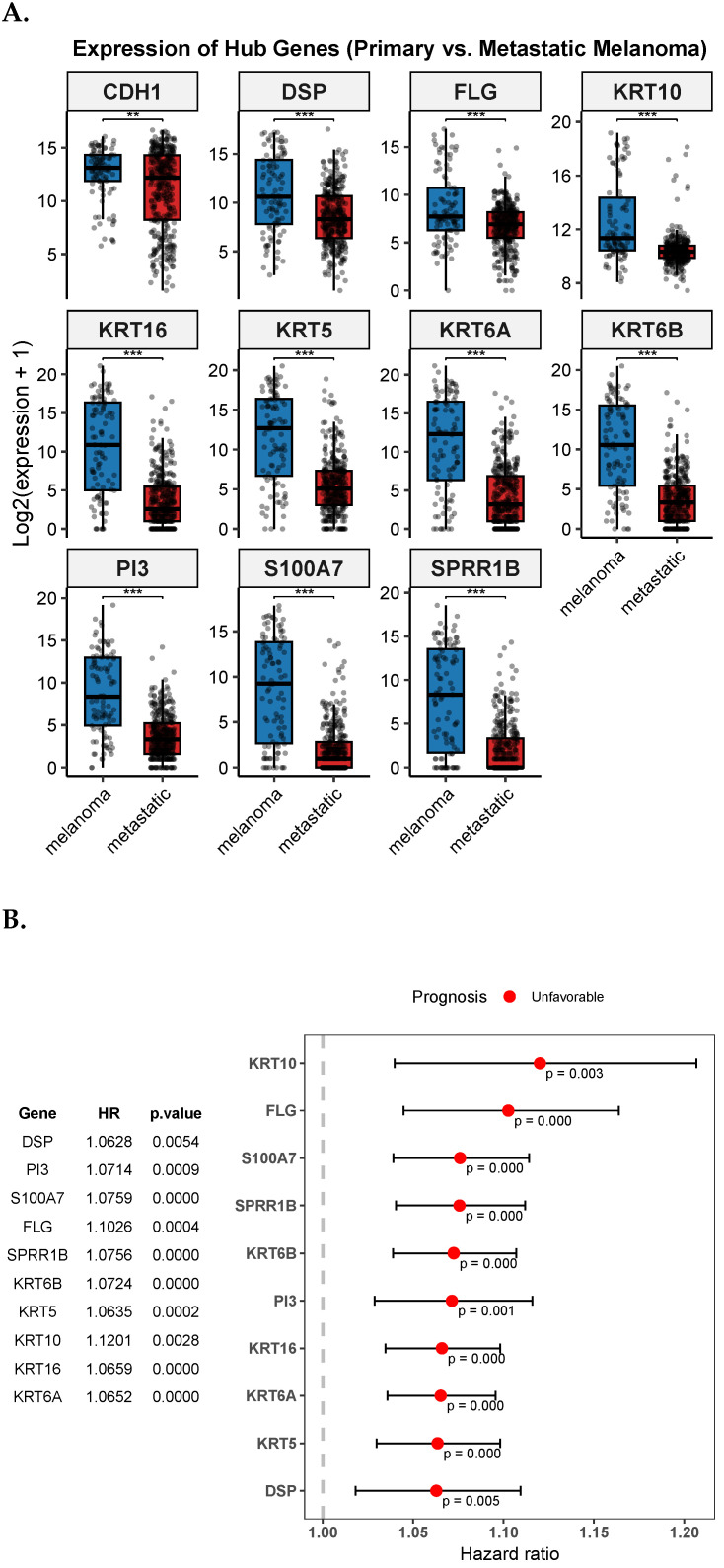
Validation of expression levels and prognostic relevance of hub genes. (**A**). Comparison of mRNA expression levels between primary (*n* = 103) and metastatic melanoma tissues (*n* = 367) based on the TCGA-SKCM cohort. Asterisks indicate statistical significance from Wilcoxon rank-sum test: *p* < 0.01 (**), *p* < 0.001 (***). (**B**). Univariate survival analysis was performed using the TCGA-SKCM cohort. Hub genes were classified as unfavorable (HR>1) or favorable (HR<1) based on the hazard ratios.

**Figure 9 diagnostics-15-01770-f009:**
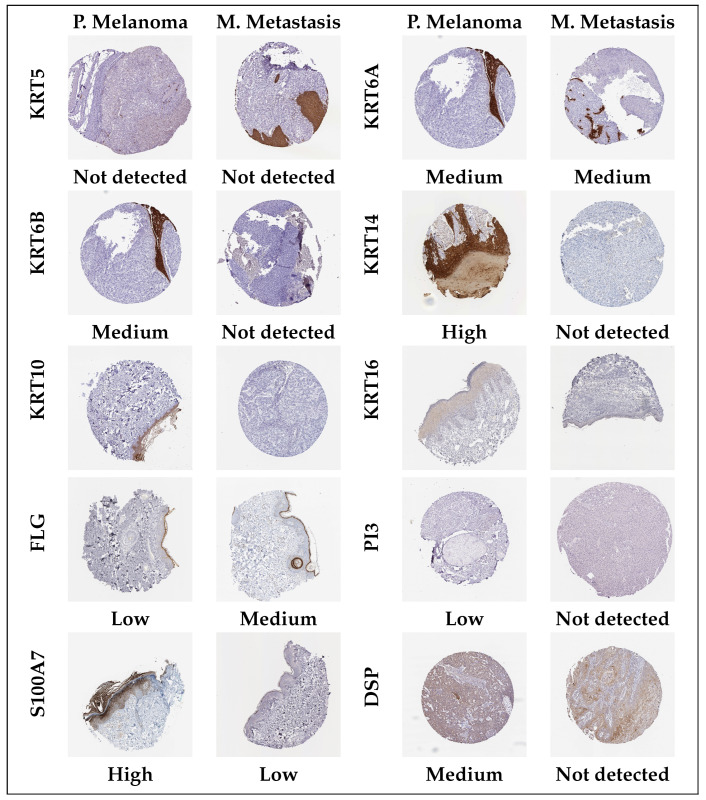
Immunohistochemistry (IHC) staining analysis of the 10 (excluding SPRR1B) hub genes in melanoma metastasis. The level of staining was grouped as ‘not detected’, ‘low’, ‘medium’, and ‘high’.

**Figure 10 diagnostics-15-01770-f010:**
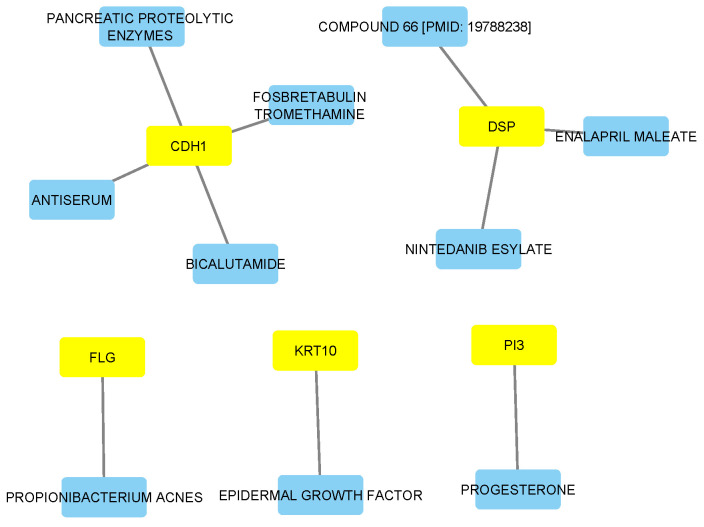
Drug–gene interaction network of hub genes in melanoma metastasis. The network shows interactions between 5 hub genes (yellow boxes) and the corresponding drugs (light blue boxes) based on DGIdb data. The network was visualized using Cytoscape.

**Table 1 diagnostics-15-01770-t001:** Core attributes of the four datasets considered in the integrated metastatic melanoma meta-analysis.

		Melanoma	
**GEO Name**	**Platform**	**Primary**	**Metastatic**
GSE7553 [23]	GPL570 (Affymetrix Human Genome U133 Plus 2.0)	46	12
GSE15605 [24]	GPL570 (Affymetrix Human Genome U133 Plus 2.0)	31	52
GSE46517 [25]	GPL96 (Affymetrix Human Genome U133A)	31	73
GSE8401 [26]	GPL96 (Affymetrix Human Genome U133A)	14	40

**Table 2 diagnostics-15-01770-t002:** Summary of drug–gene interactions retrieved from DGIdb database for the identified hub genes in melanoma metastasis.

Gene	Drug	Approval Status	Indication	Interaction Score
*PI3*	PROGESTERONE	Approved	Pre-term birth risk reduction	1.77
*KRT10*	EPIDERMAL GROWTH FACTOR	Not Approved	Not available	11.60
*DSP*	COMPOUND 66 [PMID: 19788238]	Not Approved	Not available	1.58
*DSP*	ENALAPRIL MALEATE	Approved	Antihypertensive agent	1.83
*DSP*	NINTEDANIB ESYLATE	Approved	Antineoplastic agent	1.39
*CDH1*	FOSBRETABULIN TROMETHAMINE	Not Approved	Not available	13.05
*CDH1*	ANTISERUM	Not Approved	Not available	0.75
*CDH1*	BICALUTAMIDE	Approved	Antineoplastic agent	1.31
*CDH1*	PANCREATIC PROTEOLYTIC ENZYMES	Not Approved	Not available	6.53
*FLG*	PROPIONIBACTERIUM ACNES	Not Approved	Not available	52.20

**Table 3 diagnostics-15-01770-t003:** Structural druggability evaluation of hub genes using the AlphaFold-predicted structures and ProteinPlus (DoGSiteScorer). A druggability score above 0.6 indicates high-confidence targets for structure-based drug design.

Pockets	Gene	Druggability_Score	Volume
P 0	*CDH1*	0.80	1903.56
P 0	*DSP*	0.82	792.39
P 0	*FLG*	0.80	3880.47
P 0	*PI3*	0.84	760.04
P 0	*S100A7*	0.76	615.68
P 0	*SPRR1B*	0.26	109.65
P 0	*KRT5*	0.58	113.20
P 0	*KRT6A*	0.71	374.28
P 0	*KRT6B*	0.78	1058.92
P 0	*KRT10*	0.60	396.39
P 0	*KRT16*	0.80	1059.57

## Data Availability

This study analyzed datasets that are publicly accessible. The data is accessible at the following links: GSE7553 (https://www.ncbi.nlm.nih.gov/geo/query/acc.cgi?acc=GSE7553, accessed on 10 May 2025), GSE15605 (https://www.ncbi.nlm.nih.gov/geo/query/acc.cgi?acc=GSE15605, accessed on 10 May 2025), GSE46517 (https://www.ncbi.nlm.nih.gov/geo/query/acc.cgi?acc=GSE46517, accessed on 10 May 2025), GSE8401 (https://www.ncbi.nlm.nih.gov/geo/query/acc.cgi?acc=GSE8401, accessed on 10 May 2025), and TCGA-SKCM (https://portal.gdc.cancer.gov/projects/TCGA-SKCM, accessed on 8 January 2024). The codes for the analysis are available from the corresponding author upon reasonable request.

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
