# Peer review of "Integrated Meta-Analysis Identifies Keratin Family Genes and Associated Genes as Key Biomarkers and Therapeutic Targets in Metastatic Cutaneous Melanoma"

_diagnostics, 2025, doi:10.3390/diagnostics15141770_

Round 1

Reviewer 1 Report

Comments and Suggestions for Authors

The paper, "Integrated Meta-Analysis Identifies Keratin Family Genes and Associated Genes as Key Biomarkers and Therapeutic Targets in Metastatic Skin Cutaneous Melanoma", effectively summarizes the study's focus on keratin-related genes in melanoma metastasis. However, the following refinements could enhance clarity and impact:

The abstract highlights immune modulation (e.g., KRT5 and CD8+ T cells). Should the title explicitly mention "immune infiltration" or "immunomodulatory potential"?

Introduction

  1. The introduction outlines melanoma’s aggressiveness and rising incidence but could better articulate the unresolved questions about keratin genes in metastasis. How does this study advance beyond prior work (e.g., Li et al., 2021, on keratinocyte differentiation)?

  2. The authors should cite recent literature (e.g., , https://doi.org/10.1007/s12626-025-00181-x   and https://doi.org/10.1007/s12672-025-02279-8 ) to contextualize their focus on keratin genes and immune infiltration.

Literature Review

The discussion of keratin genes (KRT5, KRT14) lacks references to their roles in other cancers (e.g., Song et al., 2022, on KRT68 in bladder cancer). Could the authors compare their findings to these studies?

The link between KRT5 and CD8+ T cells is intriguing but would benefit from citing https://doi.org/10.3390/diagnostics13132264 on immune biomarkers in melanoma.

Methods

Why were these four GEO datasets chosen? Were batch effects or platform differences (GPL570 vs. GPL96) addressed?

The authors used DExMA and RobustRankAggreg. Were results cross-validated with other tools (e.g., INMEX)?

Five methods (MCC, Betweenness, etc.) were used. Were weights assigned, or were all treated equally?

Results

The GO/KEGG terms (e.g., "epidermis development") are keratinocyte-centric. How do these pathways relate to melanocyte-derived melanoma?

Ten hub genes were identified, but only five (KRT5, KRT6A/B, KRT14, IVL) correlated with survival. Why were the others retained?

KRT5’s negative correlation with tumor purity suggests stromal influence. Did the authors validate this with single-cell RNA-seq or IHC?

Discussion

The drug-gene interaction analysis yielded limited actionable targets (e.g., progesterone for PI3). How might DSP or IVL inhibitors be repurposed?

The lack of single-cell RNA-seq is noted. Could spatial transcriptomics resolve tumor-immune spatial relationships?

How do the keratin family findings align with https://doi.org/10.1007/s12626-025-00181-x on desmosomal genes in metastasis?

Conclusion

The abstract suggests KRT5 as an immunotherapeutic target. Does the data support mechanistic insights (e.g., PD-L1 crosstalk) or only correlation?

Should the authors propose experimental validation (e.g., CRISPR knockdown of KRT5 in melanoma cell lines)?

Author Response

Kindly see the attached PDF for our humble responses to your concerns.

Thank you.

Reviewer 2 Report

Comments and Suggestions for Authors

In this manuscript, Sumaila Abubakari and colleagues present a meta-analysis of differentially expressed genes in primary versus metastatic melanoma, with a particular focus on keratin family genes. The authors used several publicly available datasets to derive 10 hub genes potentially associated with melanoma progression. While the study addresses an important topic and leverages integrative data approaches, several critical issues regarding dataset integrity, methodology, and interpretation must be addressed to enhance the veracity and relevance of the findings.

Major Comments:

  • The identification of keratin family genes (e.g., KRT5, KRT6A/B, KRT14) as metastasis-associated biomarkers is potentially relevant, but the novelty of this finding is limited. Keratin dysregulation has already been implicated in tumor progression across various cancers, including melanoma. The authors should provide clarification on the novelty of their findings in the context of what is already known. 
    A major concern is that the authors did not adequately exclude non-cutaneous melanoma (SKCM) samples. Specifically, the GSE7553 dataset appears to include basal cell carcinoma and squamous cell carcinoma samples, not just melanoma. This is problematic, as keratin genes like KRT5, KRT6A, and KRT14 are canonical epithelial markers, not typically expressed in melanocytic lineage cells. The authors must confirm that all samples included are SKCM, provide this information in a supplementary table, and re-analyze the data accordingly. In addition, the sample numbers for the GSE8401 dataset appear incorrect. The authors report 42 non-metastatic and 40 metastatic samples, but the dataset contains 31 primary and 52 metastatic melanoma samples. This discrepancy should be corrected to avoid misleading conclusions.
  • The study exclusively uses microarray-based GEO datasets (Affymetrix platforms) for DEG analysis, while relying on RNA-seq data from TCGA-SKCM for validation. This introduces significant bias, as microarray and RNA-seq platforms differ in sensitivity and resolution. The authors do not discuss how these differences might affect cross-platform validation or result interpretation.
    Additionally, the manuscript lacks essential methodological details:
    - No description is provided of batch effect correction.
    - It is unclear whether quality control and normalization were applied consistently across all datasets.
    - There is no mention of tumor purity filtering.
  • The authors state that both univariate and multivariable Cox regression analyses were performed using TCGA-SKCM clinical data. However, they do not list the covariates included in the multivariable model (e.g., age, sex, tumor stage, mutation status), nor do they present any multivariable results. It is impossible to assess whether the reported prognostic associations are independent of known clinical confounders. Full Cox model details (hazard ratios, confidence intervals, p-values, and included variables) must be reported, ideally in a supplementary table.
    The drug-gene interaction analysis is superficial. 
    The claim that KRT5 may serve as an immunotherapeutic target is speculative and unsupported by experimental data. More robust immune deconvolution methods (e.g., CIBERSORT, xCell) could lend stronger support to the reported associations.
    Although the manuscript acknowledges limitations, this section is too brief. The authors should elaborate on the constraints of bulk transcriptomic data, especially given the known heterogeneity and plasticity of melanoma. They should also discuss how melanoma subtypes might influence gene expression profiles and potentially confound the DEG analysis.

Author Response

Kindly see the PDF attached as our humble responses to your concerns. Thank you.

Round 2

Reviewer 2 Report

Comments and Suggestions for Authors

I appreciate the authors’ comprehensive responses to my comments. The revised manuscript has improved significantly in both clarity and scientific rigor. Most of the major concerns have been adequately addressed.